# The Effect of Supplementation with Betaine and Zinc on In Vitro Large Intestinal Fermentation in Iberian Pigs under Heat Stress

**DOI:** 10.3390/ani13061102

**Published:** 2023-03-20

**Authors:** Zaira Pardo, Iván Mateos, Cristina Saro, Rómulo Campos, Héctor Argüello, Manuel Lachica, María José Ranilla, Ignacio Fernández-Fígares

**Affiliations:** 1Departamento de Nutrición y Producción Animal Sostenible, Estación Experimental del Zaidín, Consejo Superior de Investigaciones Científicas, (CSIC) Profesor Albareda 1, 18008 Granada, Spain; 2Departamento de Producción Animal, Universidad de León, Campus de Vegazana s/n, 24071 León, Spain; 3Instituto de Ganadería de Montaña, CSIC-Universidad de León, Finca Marzanas s/n, Grulleros, 24346 León, Spain; 4Departamento de Ciencia Animal, Universidad Nacional de Colombia, Carrera 32 # 12-00, Palmira 76531, Colombia; 5Departamento de Sanidad Animal, Universidad de León, Campus de Vegazana s/n, 24071 León, Spain

**Keywords:** Iberian pig, heat stress, zinc, betaine, *in vitro* hindgut fermentation, short-chain fatty acids

## Abstract

**Simple Summary:**

Heat stress has negative consequences in animal husbandry and pigs are highly susceptible because of their scattered sweat glands and elevated metabolic rate. Above 25 °C, pigs trigger mechanisms such as diverting blood from their internal organs to skin or decreasing the feed intake, which can affect the intestinal microbiota and capacity of fermentation. Betaine and zinc have been used as nutritional mitigation strategies to alleviate the effects of heat stress in pigs, but their effects on hindgut fermentation are unknown. By using an *in vitro* fermentation system and employing as the inoculum the rectal content from heat-stressed Iberian pigs supplemented or not with betaine or zinc, we showed that both supplements modified the pattern of hindgut fermentation in heat stressed pigs, having beneficial effects on acetate and butyrate production. The abundance of some bacterial groups was also modified, indicating an effect on the microbial structure. Because of the intricate interactions between the host and gut microbiota, additional studies are necessary to confirm these results under *in vivo* conditions.

**Abstract:**

We investigated the effects of betaine and zinc on the *in vitro* fermentation of pigs under heat stress (HS). Twenty-four Iberian pigs (43.4 ± 1.2 kg) under HS (30 °C) were assigned to treatments for 4 weeks: control (unsupplemented), betaine (5 g/kg), and zinc (0.120 g/kg) supplemented diet. Rectal content was used as the inoculum in 24-hincubations with pure substrates (starch, pectin, inulin, cellulose). Total gas, short-chain fatty acid (SCFA), and methane production and ammonia concentration were measured. The abundance of total bacteria and several bacterial groups was assessed. Betaine increased the acetate production with pectin and inulin, butyrate production with starch and inulin, and ammonia concentration, and decreased propionate production with pectin and inulin. The abundance of *Bifidobacterium* and two groups of *Clostridium* decreased with betaine supplementation. Zinc decreased the production of SCFA and gas with starch and inulin, associated with diminished bacterial activity. Propionate production decreased with starch, pectin, and inulin while butyrate production increased with inulin, and isoacid production increased with cellulose and inulin in pigs supplemented with zinc. The ammonia concentration increased for all substrates. The *Clostridium* cluster XIV abundance decreased in pigs fed zinc supplemented diets. The results reported were dependent on the substrate fermented, but the augmented butyrate production with both betaine and zinc could be of benefit for the host.

## 1. Introduction

High environmental temperatures cause heat stress (HS) with negative consequences in animal husbandry around the world [1]. Pigs are highly susceptible to HS because of their scattered sweat glands and elevated metabolic rate [2]. The negative impact of HS on pig growth has been shown in cosmopolitan pigs [3] and also in purebred Iberian pigs [4]. The exact mechanisms of heat stress-induced reactions in productivity are unknown but might be partially mediated by its effects on intestinal integrity. Heat stress decreased intestinal integrity and increased endotoxin permeability in growing crossbred gilts [5,6], while the digestibility of gross energy, dry matter, crude protein, and ash decreased in finishing crossbred barrows after constant exposure to heat stress compared to the pair fed thermoneutral pigs [7]. Alterations to the pig microbiota in HS conditions have been reported [8,9], which may lead to changes in the intestinal fermentation pattern. Microbiota present in the intestine carry out the fermentation of dietary fiber, generating gases and short chain fatty acids (SCFA) [10]. Indeed, we have shown increased *in vitro* intestinal fermentation in Iberian pigs after HS [11]. Unsurprisingly, zinc [12,13] and betaine [14,15] have been used as nutritional mitigation strategies to alleviate HS in pigs. Zinc attenuated heat stress-induced changes in ileum integrity in crossbred gilts [16], showing a meaningful role in maintaining gastrointestinal tract barrier functionality [17]. Betaine is a compatible osmolyte [18] that is able to prevent dehydration and possibly improve digestibility [19].

To the best of our knowledge, there is no information as to whether or not zinc and betaine may affect hindgut fermentation when pigs are exposed to elevated temperatures. We hypothesized that betaine and zinc may alter the microbiota composition and fermentation pattern under heat stress. The aim of the present research was to further study the effects of dietary betaine and zinc on the hindgut fermentation of pigs subjected to HS.

## 2. Materials and Methods

### 2.1. Animals, Treatments, and Diets

Animal procedures and care were in accordance with the European Union guidelines (Directive 2010/63/EU of the European Parliament and of the Council on the protection of animals used for scientific purposes) and Spanish Ministry of Agriculture rules (RD53/2013). The experimental protocols used have been previously approved by the Bioethical Committee of the Spanish Council for Scientific Research (CSIC, Madrid, Spain) and the competent local authority (Junta de Andalucia, Spain, project authorization 28/06/2016/118).

The experiment was performed with twenty-four pure Iberian (Sánchez Romero Carvajal strain) barrows (43.4 ± 1.2 kg) that originated from an antibiotic-free herd supplied by Sanchez Romero Carvajal (Puerto de Santa María, Cádiz, Spain), housed in a temperature-controlled room with no exposure to antibiotics during the whole process of this study and allotted into individual slatted pens (2 × 1 m), allowing for visual contact among them. On arrival, the pigs were fed a barley-corn-soybean meal diet supplemented with amino acids to obtain an optimal amino acid profile (NRC, 2012) and cover all of the other nutrient requirements [20]. Pigs were kept for one week in thermoneutral conditions (20 ± 1 °C) to adapt to the basal diet and the facilities. Three diets were formulated: (i) the control diet (Ctrl treatment, Appendix A); (ii) control diet supplemented with 0.120 g/kg Zn sulfate, (Zn treatment, ZnSO_4_·H_2_O, 98%, VWR, Leuven, Belgium); and (iii) the control diet supplemented with 5 g/kg betaine (Bet treatment; TNI-Betain, natural origin, anhydrous, 96%, Trouw Nutrition-Nutreco; Madrid, Spain). Following an adaptation period of one week, pigs were assigned to the experimental diets and fed ad libitum for four weeks under constant heat stress conditions (30 ± 1 °C). The temperature was progressively raised (from 20 to 30 °C) and kept constant during day and night using an air conditioning apparatus (LG UM36, LG Electronics Inc., Changwon, Republic of Korea). Relative humidity and temperature inside the room were monitored with the aid of a data logger (HOBO UX100-011; Onset Computer Corporation, Bourne, MA, USA) set to record values every 30 min. The experiment was carried out in four replicates, each with two pigs per treatment. Water was freely available. Pigs were slaughtered (59.7 ± 1.5 kg) by exsanguination after electrical stunning and a sample of the rectum content was collected under anaerobic conditions (CO_2_-filled container) and immediately frozen and kept at −20 °C until the *in vitro* incubation trial and DNA extraction. Further details about the *in vivo* experiments may be found in [4].

### 2.2. Substrates and In Vitro Incubations

The substrates used in this experiment differed in the composition of sugars and in the linkages between sugars, which can affect fermentation. A mix of three starches (denominated as starch in the text) of 2:2:1 corn starch (SIGMA S-4126): potato starch (SIGMA S-2004): wheat starch (SIGMA S-2760), pectin from citrus peel (SIGMA P-9135), inulin from chicory (SIGMA I-2255-25G), and microcrystalline cellulose (Merck 1-02331.0500) were used. Substrates were dried overnight at 40 °C, left to stand at room temperature in a desiccator, and 300 mg was weighed into glass bottles (120 mL).

On the day of incubation, the rectum content was thawed at 5 °C and pooled every two pigs of each experimental treatment. The experiment was carried out in four runs where bottles of substrates were incubated in duplicate with inoculum prepared from the rectum content of each of the three treatments (Ctrl, Bet, and Zn). Blanks (bottle with inoculum but without substrate) were used (two/per inoculum and run) to correct the gas production of the substrates and to detect abnormalities between the runs.

The inoculum was prepared by mixing 25 g of thawed rectum content and 500 mL of the buffered anaerobic culture medium of Goering and Van Soest [21] without trypticase, (5% wt/vol final concentration of rectum content). The mixture was homogenized in a Stomacher (model number BA6021, Seward Medical, London, UK) at 230 rpm for sixty seconds, filtered through a nylon bag (200 µm mesh screen), and added (30 mL) into each bottle under CO_2_ flushing. Finally, the bottles were sealed with rubber stoppers and aluminum caps and incubated (39 °C) for 24 h.

### 2.3. Analysis of Samples

Gas production was measured with a pressure transducer (Delta Ohm DTP704-2BGI, Herter Instruments SL, Barcelona, Spain) and a calibrated syringe after 24 h of incubation. Evacuated tubes (Terumo Europe N. V., Leuven, Belgium) were used to store 10 mL of gas samples for methane analysis. Then, the bottles were opened and the fermentation process was stopped by introducing the bottles in ice water. A sample was collected (4 mL) from each bottle, mixed with 100 µL of 20% sulfuric acid to properly conserve the sample and stored at −20 °C until analysis for short chain fatty acids (SCFAs) and ammonia.

#### 2.3.1. Short Chain Fatty Acid Analysis

Liquid samples taken from the incubation bottles were manipulated as described by Saro et al. [22]. Briefly, 800 μL of the content was added to 500 μL deproteinizing solution (metaphosphoric acid (2%) and crotonic acid (0.06%) as the internal standard for SCFA determinations). Short chain fatty acids were analyzed using a gas chromatograph (Shimadzu GC-2010. Duisburg, Germany). The amount of SCFA produced was obtained by subtracting the quantity initially present.

#### 2.3.2. Ammonia and Methane Analysis

Liquid samples were thawed and centrifuged and the supernatant was analyzed to determine the concentration of ammonia through a colorimetric method [23].

The methane contained in the gas samples was analyzed in a chromatograph following the method described by Martinez [24]. In brief, 2 mL of gas was injected into a gas chromatograph (Shimadzu GC 14B; Shimadzu Europa GmbH, Duisburg, Germany), heated at 200 °C, and equipped with a flame ionization detector and a column packed with Carboxen 1000 (Supelco, Madrid, Spain). Helium was used as the carrier gas and peaks were identified by a comparison with a standard of known methane concentration.

### 2.4. DNA Extraction and Quantitative Polymerase Chain Reaction

The total genomic DNA was extracted from the rectum content thawed samples (100 mg) of each pig using a commercial kit (QIAamp DNA Stool Mini kit, QIAgen, Valencia, CA, USA) following the procedure of Yu and Morrison [25], except that the samples were treated with cetyltrimethylammonium bromide to remove the PCR inhibitors. A Nanodrop ND-1000 (Nano-Drop Technologies, Wilmington, DE, USA) was used to measure the absorbance ratios (A_260_:A_280_ and A_260_:A_230_) of the eluted DNA. The absorbance ratio (A_260_:A_280_) showed an average value of around 1.8, generally accepted as pure DNA and the ratio (A_260_:A_230_) was within the range 2.0–2.2 [26].

Eluted DNA (2 µL) was used to assess the abundance of total bacteria and six bacterial groups by quantitative PCR (qPCR). The specific primer sequences for each bacterial group (total bacteria, *Lactobacillus*, *Bifidobacterium*, Enterobacteriaceae, *Bacteroides*, and the *Clostridium* cluster IV and XIV) are shown ([27] Appendix A). Thermocycling was carried out in a QuantStudio 1 Thermal Cycler (Applied Biosystem, Foster City, CA, USA) and the cycling conditions were 95 °C for 10 min (initial cycle of denaturation), followed by 40 cycles of 95 °C for 15 s, and 60 °C for 60 s. To determine the specificity of amplification, analysis of the product melting was carried out after each amplification. Each PCR reaction mixture contained 8 µL SYBR Green PCR Master Mix (Applied Biosystems, Warrington, UK), 0.3 µL of 10 µM each primer, 0.12 µL of Rox (1:10. Thermo Scientific, Vilnius, Lithuania), 9.28 µL of DNase-free water, and extracted DNA. For qPCR quantification of the different bacterial groups, the bacterial type strains were used to prepare the calibration standards using cultures of the different species at specified concentrations (Appendix A). The DNA of these cultures was extracted as described for fecal samples. For each bacterial group, serial dilutions of the DNA obtained were run in each qPCR plate together with the fecal samples.

### 2.5. Statistical Analysis

This experiment was designed as a complete crossover design with three treatments and four replicate runs. Data were analyzed separately for each substrate. Prior to the analysis of variance, homogeneity between the variances and normality among treatments were confirmed using the Bartlett and Ryan–Joiner tests, respectively. To compare the differences between treatments for each substrate, data were subjected to ANOVA using the MIXED procedure of SAS (Version 9.3, SAS Inst. Inc., Cary, NC, USA). The fixed effect of treatment and the random effect of fermentation flask nested within the experimental run were included in the main model considering the fermentation flask as the experimental unit. Additionally, pre-planned contrasts were generated using the contrast statement procedure of SAS to evaluate the treatment effects (control vs. Zn, and control vs. betaine). Results were regarded as showing a tendency of significance with *p*-values between 0.05 and 0.10. Data were presented as the mean ± standard error, *n* = 8.

## 3. Results

### 3.1. Effect of Dietary Betaine on In Vitro Intestinal Fermentation

The gas production and fermentation parameters of different substrates after 24 h *in vitro* fermentation by fecal inocula obtained from pigs under heat stress conditions and fed diets supplemented with betaine are shown in Table 1, Table 2, Table 3 and Table 4. 

Although gas production was not affected for any substrate (*p* > 0.05) and total SCFA production only showed a tendency to increase when pectin was fermented (*p* = 0.0778), betaine supplementation increased acetate production when pectin and inulin were fermented (6 and 16%; *p* = 0.0019 and *p* = 0.0173, respectively), but showed no effect for starch and cellulose (*p* > 0.05). Furthermore, the acetate molar proportion was augmented when inulin, pectin, and cellulose were fermented (4–16%; *p* < 0.001) and showed a trend to increase when starch was fermented (*p* = 0.0985). Betaine decreased the propionate production when pectin and inulin were fermented (11.6 and 30.4%; *p* = 0.0249 and *p* = 0.0002, respectively). Additionally, betaine diminished the propionate molar proportion when starch, pectin, and inulin were used as substrates (18–35%; *p* < 0.001).

Butyrate production increased with the addition of betaine when starch and inulin were incubated (12.5 and 39%; *p* = 0.0512 and *p* = 0.0029, respectively), but decreased with pectin (11%; *p* = 0.0143). Additionally, betaine increased the butyrate molar proportion when starch, inulin, and cellulose were fermented (20–31%; 0.05 < *p* < 0.001), although it decreased when pectin was used as the substrate (16%; *p* < 0.001).

Betaine did not affect the isoacid production except for an increase when the cellulose was fermented (25.8%; *p* = 0.0005). Furthermore, the isoacid molar proportion increased when starch and inulin (53–57%; *p* < 0.05) were incubated. Valerate production and the molar proportion increased when starch and pectin were fermented (72–80%; (0.001 < *p* < 0.05) and 69–106%; (*p* < 0.001), respectively), but decreased with inulin or cellulose (16–75%; *p* < 0.001 and 60–70%; *p* < 0.001, respectively). Betaine did not affect the methane production for any substrate (*p* > 0.05). However, betaine increased the ammonia concentration for all substrates (5–10%; 0.01 < *p* < 0.05).

Abundances of the bacterial populations were affected by the treatments (Table 5). Betaine was associated with a decrease in *Bifidobacterium* and *Clostridium* cluster XIV (3.2 and 5.6%, respectively). Additionally, *Clostridium* cluster IV tended to decrease with betaine supplementation (0.5%; *p* = 0.0541). 

### 3.2. Effects of Dietary Zn on In Vitro Intestinal Fermentation

Gas production and fermentation parameters of different substrates after 24 h *in vitro* fermentation by fecal inocula obtained from pigs under heat stress conditions and fed diets supplemented with zinc are shown in Table 1, Table 2, Table 3 and Table 4. Zinc supplementation decreased the production of SCFA when starch and inulin were fermented (15–23%; 0.05 < *p* < 0.001). Additionally, gas production decreased for starch and inulin (4–5%; 0.05 < *p* < 0.01). Zinc supplementation diminished acetate production for starch (15%; *p* = 0.0154), but increased when cellulose was incubated (73%; *p* = 0.0002). The acetate molar proportion increased when inulin and cellulose were the substrates of fermentation (10–25%; 0.01 < *p* < 0.001). Zn decreased the propionate production and propionate molar proportion when starch, pectin, and inulin were fermented (24–62%; *p* < 0.001 and 12–50%; 0.01 < *p* < 0.001, respectively). However, zinc increased the butyrate production when inulin (+18%; *p* < 0.01) was used as the substrate and the butyrate molar proportion when inulin and pectin were fermented (12.5–20%; 0.05 < *p* < 0.01). On the other hand, Zn decreased the butyrate molar proportion when cellulose was fermented (7.7%; *p* < 0.05). Methane production was not responsive to Zn supplementation irrespective of the substrate fermented (*p* > 0.05). Zn increased isoacid production when cellulose and inulin were used (76–111%; *p* < 0.001), and tended to increase when pectin was fermented (45%; *p* = 0.068) and increased the isoacid molar proportion with starch, inulin, and cellulose (34–182%; 0.01 < *p* < 0.001) and tended to increase with pectin (47%; *p* = 0.064). Zn increased the valerate production when pectin was fermented (112; *p* < 0.01) but decreased when inulin and cellulose were fermented (80%; *p* < 0.001). Zn increased the valerate molar proportion when starch and pectin were fermented (54–129%; 0.001 < *p* < 0.01), but decreased when inulin or cellulose were fermented (71–80%; *p* < 0.001). Similarly, ammonia concentration increased for all substrates (5–17% *p* < 0.01). In relation to the abundance of bacterial populations (Table 5), *Clostridium* cluster XIV decreased (5%; *p* = 0.004) in pigs supplemented with Zn compared to the control diet. 

## 4. Discussion

*In vitro* gas production models have been proven to be highly efficient to determine the fermentation characteristics in the hindgut of pigs [28,29]. The study here reported was conducted in order to evaluate the fermentation of pure substrates by inocula from Iberian pigs under heat stress conditions and fed with diets supplemented with betaine or Zn. Although no information on how dietary supplementation with betaine and Zn affect fermentation under heat stress conditions exists, it has been reported that betaine supplementation under thermoneutral conditions is capable of altering the composition of the bacterial community in the gastrointestinal tract of piglets [30] and poultry [31]. Moreover, we have recently shown that betaine increases the net portal absorption of SCFAs [32] in Iberian pigs. 

The influence of heat stress on the *in vitro* fermentation characteristics is mostly dependent on the source of carbohydrate fermented (complete diet, pure substrates…), which is in agreement with the results of the present study, where cellulose showed, under heat stress conditions, a remarkably lower total gas production and SCFA concentrations than starch, inulin, or pectin. Therefore, the extent of fermentation seems to depend on both the prevalence of environmental conditions and the fermentability of the individual carbohydrates. Hence, the use of pure substrates with different characteristics allowed us to have a variety of fermentation profiles [11,33,34]. Indeed, it has been reported that the fermentation of dietary fiber by pig intestinal microbes mainly depends on its amount and solubility [35]. Previous works have demonstrated that pig feces can be used as inoculum for *in vitro* fermentation techniques, replacing the intestinal content [11,36,37] and permitting a valid study of the microbial activity present in the large intestine [38,39]. We used a 24 h incubation time to simulate the transit time of pigs fed diets based on cereals [40].

The fermentation of dietary fiber by the microbiota resulted in the production of SCFA [41] and the most important end-products were acetate, propionate, and butyrate, which are significant for intestinal health and additionally provide energy to animals. Short chain fatty acids have important functions in the gut mucosa and are absorbed by the colonic mucosa, but only acetic acid is capable of reaching systemic circulation in significant quantities via the liver [42]. In pigs, the concentration of short chain fatty acids in feces represents a proportion of the SCFA produced by gut bacteria, since a significant amount is absorbed [43]. At the same time, microbial cells and mucins present in the hindgut are rich in insoluble fractions of fiber, which causes the production of SCFA from components that are not in the diet [44].

### 4.1. Effects of Dietary Betaine on Gas and Short-Chain Fatty Acid Production, Ammonia Concentration, and Methane Production

The generation of gas is a classic indicator reflecting the degree of fermentation *in vitro* [45]. Under heat stress conditions, betaine failed to affect microbial fermentation activity *in vitro* in terms of the total gas and total SCFA production. Similarly, pure betaine sources did not affect SCFA production under conditions of osmotic stress [46] and under normal buffered conditions [39] in pigs evaluated *in vitro* through the modified Hohenheim gas test. In contrast, augmented bacterial growth of an *Enterococcus faecalis* pure culture has been reported under osmotic stress conditions after betaine addition [47]. A direct comparison of the results from *in vitro* studies is risky due to the use of completely different procedures. Furthermore, the use of cell cultures of single bacteria strains cannot take into account the complexity of a multitude of different bacteria strains present in pig feces used as microbial inoculum.

Betaine did not affect methane production under the conditions of the present experiment. Although betaine may be metabolized to methane by rumen methanogenic archaea [48], we have found no information on the effect of betaine on methane production in pigs. 

In the current study, we found an increase in acetate production with pectin and cellulose as substrates and in the acetate molar proportion when starch and inulin were fermented. This could have a positive effect on the performance of growing pigs, as acetate is absorbed in the large intestine of pigs and can be used for fat synthesis [49]. Additionally, acetate may be utilized for butyrate formation by fecal bacteria [50] (i.e, Clostridiaceae members).

Betaine supplementation decreased the propionate production and molar proportion for different substrates in the present study. In contrast, a study by Rink et al. [46] reported increased propionate production in the hindgut during hyperosmotic stress following the supplementation of a native betaine source, but not with pure betaine, in the diet of pigs. 

Butyrate production and molar proportion increased in the betaine group for different substrates. In agreement, Rink et al. [46] reported an increased butyric acid production due to supplementation with condensed molasses solubles, but not with pure betaine under thermoneutral conditions. It cannot be discarded, however, that substances other than betaine present in condensed molasses solubles stimulated fermentation activity. 

Butyrate is absorbed mainly by colonocytes, acting as a source of energy [41], accounting for 70% of all energy consumed in the colon [51]. Moreover, it is capable of regulating the growth of intestinal cells, leading to an improvement in intestinal proliferation [52] as well as regulating the absorptive capacities of the small intestine in pigs [53]. Furthermore, butyrate has shown additional beneficial effects such as improving barrier function [54] and having anti-inflammatory [55] and anti-tumorigenic properties [56,57].

Betaine did not affect the production of isoacids and molar proportion, except when cellulose was fermented. On the other hand, the effects on the valerate production and valerate molar proportions depended on the substrate fermented. Likewise, under osmotic stress conditions, the addition of betaine did not alter the branched chain proportion using the Hohenheim gas test [46]. Instead, betaine increased the ammonia concentration in the media, which is usually considered as an indicator for increased protein breakdown when undigested protein reaches the hindgut. However, the fermentation substrates in the present experiment were pure carbohydrates, so ammonia may only come from ammonia salts present in the media as source of N for the microbiota. Increased ammonia concentration should not be interpreted as a result of protein breakdown because no protein was added to the media. 

Although total bacteria abundance was not affected by betaine treatment, a decrease in *Bifidobacterium*, *Clostridium cluster XIV*, and *Clostridium cluster IV* was found in the present study. Although it has been shown that *Clostridium* could ferment polysaccharides to SCFAs [58], we were not able to detect a decrease in the gas production or total SCFA in pigs fed the betaine supplemented diets under heat stress conditions. Furthermore, although betaine led to methanogenic activity in members of the genera Clostridium as the first step in betaine degradation [59,60], the increased abundance reported did not translate to changes to methane production in the present experiment. As a matter of fact, Schrama [61] showed that dietary betaine increased the methane production over time in pigs. The decreased abundance of *Bifidobacterium*, beneficial bacteria for the host, in the betaine supplemented pigs was not expected. In addition, under *in vivo* conditions, conflicting results exist on the effect of supplementary betaine on the intestinal microbial fermentation pattern in pigs [19,62].

In summary, although betaine did not affect the overall total SCFA or gas production under heat stress conditions, the fermentation profile was affected so that the increase in acetate and butyrate may lead to a trophic effect on the pig’s intestinal epithelium. Changes to the abundance of selected bacterial populations were also observed. 

### 4.2. Effects of Dietary Zn on Short-Chain Fatty Acids, Gas Production, Ammonia Concentration, and Methane Production

Starting in June 2022, the European Union has banned the use of zinc for pharmacological use in pig diets [63]. Therefore, the amount of zinc used to supplement the Zn diet (120 mg/kg) was below the maximum permitted dose rate of 150 ppm total dietary zinc (Commission Implementing Decision of 26.6.2017, C (2017) 4, 529 Final) as a feed additive.

In the present study, dietary Zn supplementation under heat stress conditions decreased the *in vitro* total SCFA production when starch and inulin were fermented, in parallel with decreased gas production with starch, which could be the result of an inhibition of bacterial growth by this mineral [64]. Indeed, Zn decreased the acetate production when inulin and starch were fermented. In agreement, weaned piglets fed diets supplemented with a high concentration of Zn under thermoneutral conditions showed decreased colon SCFA at different sampling dates (32, 39, 46, and 53 days of age;) [65]. However, Pieper [66] reported that ZnO (150 mg/kg) under thermoneutral conditions increased the total SCFA and acetate concentration at the terminal ileum of piglets, but no change was observed with greater concentrations (250–2500 mg/kg). 

Furthermore, a decrease in the propionate production and propionate molar ratio when starch, pectin, and inulin were fermented was found in the present experiment. Therefore, a diminished amount of propionate reaching the liver would be expected, and hence decreased gluconeogenesis from propionate, which would agree with the lower glycemia in pigs fed Zn supplemented diets [4]. *In vivo*, while Starke et al. [65] found a decrease in propionate with high Zn concentration (2425 g/kg ZnO) in the large intestine of piglets after a five-week experimental period under thermoneutral conditions, Pieper et al. [66] did not find differences after an experimental period of two weeks under thermoneutral conditions.

Interestingly, we found increased butyrate production with most substrates, which is in accordance with Pieper [66], who reported a marked increase in the ileal concentrations of butyrate when Zn was supplemented at 150 mg/kg compared with 50 mg/kg, and decreased with further zinc concentrations up to 2500 mg/kg. Butyrate has been linked to improved intestinal health [67] by supporting colonic mucosal health [68] as it is the most important energy source for enterocytes [41,69]. As previously mentioned for betaine, Zn supplementation could also alleviate part of the challenges imposed by heat stress at the intestinal level by increasing the butyrate production. 

The increased isoacid production by Zn could indicate fermentation of the protein in the hindgut, as isobutyrate and valerate are produced from branched chain amino acids [70] and is an indicator of protein fermentation [71]. However, there was no protein in the incubation media.

Valerate can be produced via the reduction of proline to 5-aminovalerate through the Stickland reaction [72] or through chain elongation of the propionate with ethanol, both by the *Clostridium* species [73]. In the present study, the effect of Zn on valerate production depended on the substrate fermented.

Zinc supplementation increased ammonia in the media with all substrates of fermentation, which may indicate decreased N incorporation into the bacterial mass. This would be in line with reduced SCFA production. 

The decrease in the abundance of the *Clostridium* cluster XIV group with Zn supplementation is in contrast with the *in vivo* results under thermoneutral conditions where weanling piglets fed increasing levels of ZnO (50, 150, 250, 1000 and 2500 mg zinc/kg) had decreased clostridial cluster XIVa at the ileum only at the highest ZnO level [66], and similarly, lactobacilli, bifidobacterial and enterococci were not affected. Nonetheless, many studies have reported that pharmacological supplementation with Zn in weaning diets could improve the intestinal microbiota [65,74,75]. Interestingly, the abundance of the *Clostridium* cluster XIV group decreased with Zn supplementation.

The reduction in fecal SCFAs has been associated with changes in the abundance of several SCFA-producing bacteria induced by heat stress such as Clostridiales, belonging to the Bacillota phylumm and Bacteroidales from Bacteroidota phylum [9,76]. Indeed, the total SCFA production and the abundance of Clostridium cluster XIV were decreased in the present experiment as a consequence of Zn supplementation. Although *Clostridium* cluster XIV bacteria utilize lactate and acetate to produce butyrate, in the conditions of the present experiment, *Clostridium* cluster XIV decreased while the butyrate production increased. It cannot be excluded, though, that other butyrate producing bacteria are responsible for the overall butyrate production. There is evidence of antimicrobial activity of Zn *in vitro* [77], and it is considered that the mode of action of high levels of Zn on pig performance is due to a modification of the intestinal microbiome [64,66,78]. However, total bacterial abundance in the present experiment was not affected by Zn supplementation, although it should be mentioned that the dose used was far below the pharmacological levels used for treating post weaning diarrhea (>1000 mg/kg).

## 5. Conclusions

The present study provides evidence of the influence of dietary betaine and Zn on large intestinal microbial metabolism under conditions of heat stress. Both dietary betaine and Zn had beneficial effects on acetate and butyrate. The lower concentration of SCFA by Zn should be discussed critically regarding their value for the host. Our findings may help to better understand the influence of betaine and zinc on the fermentation characteristics of different pure substrates and the interaction between the additives and intestinal microbiota, which may provide new perspectives for the formulation of nutrition intervention programs under heat stress conditions. As *in vitro* conditions may not reflect the complex interactions between the animal and the microbiota, further studies are needed to verify the present findings under *in vivo* conditions.

## Figures and Tables

**Table 1 animals-13-01102-t001:** Production of SCFA, molar proportions of SCFA, gas and methane production, and ammonia concentration after 24 h of the *in vitro* fermentation of starch by fecal inocula obtained from pigs under heat stress conditions and fed diets supplemented or not with betaine (5 g/kg) or zinc (0.120 g/kg). Means ± standard deviations are shown.

STARCH	Treatment			*p*-Value	
	Control	Betaine	Zinc	Bet vs. Control	Zn vs. Control
SCFA production:					
Total SCFA (µmol/g)	8835 ± 746	8539 ± 1457	7474 ± 687	0.587	0.012
Acetate	4619 ± 455	4711 ± 943	4014 ± 347	0.774	0.015
Propionate	2496 ± 421	1873 ± 464	1832 ± 328	0.002	<0.001
Butyrate	1669 ± 117	1877 ± 124	1546 ± 141	0.051	0.325
Isob + Isov	42 ± 10	61 ± 15	71 ± 22	0.211	0.085
Valerate	9 ± 2.9	16 ± 2.8	11 ± 1.8	0.022	0.311
Isoacids ^1^	51 ± 12	77 ± 16	81.7 ± 22	0.140	0.089
Molar proportions (mol/mol SCFA):					
Acetate	0.52 ± 0.02	0.53 ± 0.02	0.54 ± 0.01	0.099	0.088
Propionate	0.26 ± 0.03	0.20 ± 0.03	0.23 ± 0.02	<0.001	0.002
Butyrate	0.21 ± 0.03	0.26 ± 0.04	0.22 ± 0.03	<0.001	0.112
Isob + Isov	0.005 ± 0.001	0.007 ± 0.001	0.009 ± 0.001	0.045	0.008
Valerate	0.0009 ± 0.0003	0.002 ± 0.0003	0.001 ± 0.0002	<0.001	<0.001
Isoacids ^1^	0.006 ± 0.001	0.009 ± 0.0009	0.009 ± 0.001	0.016	0.003
Gas production (µmol/g)	16,247 ± 415	16,232 ± 663	15,441 ± 650	0.94	0.003
Methane production (µmol/g)	20.1 ± 1.31	20.2 ± 1.64	22.5 ± 0.87	0.97	0.150
Ammonia (mg N/L)	127 ± 9.0	139 ± 6.8	149 ± 9.4	0.003	<0.001

^1^ Isobutyrate, isovalerate and valerate.

**Table 2 animals-13-01102-t002:** Production of SCFA, molar proportions of SCFA, gas and methane production, and ammonia concentration after 24 h *in vitro* fermentation of pectin by fecal inocula obtained from pigs under heat stress conditions and fed diets supplemented or not with betaine (5 g/kg) or zinc (0.120 g/kg). Means ± standard deviations are shown.

PECTIN	Treatment			*p*-Value	
	Control	Betaine	Zinc	Bet vs. Control	Zn vs. Control
SCFA production:					
Total SCFA (µmol/g)	9773 ± 302	10,039 ± 243	9341 ± 527	0.078	0.192
Acetate	7791 ± 83.5	8314 ± 130	7758 ± 337	0.002	0.853
Propionate	1074 ± 152	949 ± 152	812 ± 112	0.025	<0.001
Butyrate	844 ± 110	751 ± 87	856 ± 172	0.014	0.821
Isob + Isov	56 ± 17	67 ± 11	81 ± 23	0.333	0.068
Valerate	9 ± 2.0	15 ± 6.1	18 ± 3.1	0.002	0.005
Isoacids ^1^	64 ± 19	82 ± 13	99 ± 25	0.149	0.032
Molar proportions (mol/mol SCFA):					
Acetate	0.80 ± 0.02	0.83 ± 0.02	0.81 ± 0.03	<0.001	0.228
Propionate	0.11 ± 0.013	0.09 ± 0.013	0.09 ± 0.012	<0.001	<0.001
Butyrate	0.085 ± 0.009	0.072 ± 0.007	0.085 ± 0.02	<0.001	0.041
Isob + Isov	0.006 ± 0.001	0.006 ± 0.001	0.008 ± 0.002	0.55	0.064
Valerate	0.0008 ± 0.0001	0.001 ± 0.0005	0.002 ± 0.0003	<0.001	0.002
Isoacids ^1^	0.006 ± 0.002	0.008 ± 0.001	0.010 ± 0.002	0.031	0.027
Gas production (µmol/g)	16,016 ± 252	13,211 ± 1936	14,516 ± 1015	0.24	0.098
Methane production (µmol/g)	22.1 ± 0.50	19.7 ± 1.75	21.8 ± 1.13	0.34	0.651
Ammonia (mg N/L)	171 ± 9.4	179 ± 5.7	184 ± 12	0.016	0.005

^1^ Isobutyrate, isovalerate and valerate.

**Table 3 animals-13-01102-t003:** Production of SCFA, molar proportions of SCFA, gas and methane production, and ammonia concentration after 24 h of the *in vitro* fermentation of inulin by fecal inocula obtained from pigs under heat stress conditions and fed diets supplemented or not with betaine (5 g/kg) or zinc (0.120 g/kg). Means ± standard deviations are shown.

INULIN	Treatment			*p*-Value	
	Control	Betaine	Zinc	Bet vs. Control	Zn vs. Control
SCFA production:					
Total SCFA (µmol/g)	6958 ± 599	7131 ± 1543	5389 ± 809	0.671	<0.001
Acetate	3775 ± 197	4517 ± 887	3674 ± 543	0.017	0.507
Propionate	2529 ± 460	1760 ± 544	963 ± 194	<0.001	<0.001
Butyrate	588 ± 68	817 ± 140	695 ± 125	0.003	0.038
Isob + Isov	23 ± 5	26 ± 4	49 ± 11	0.328	<0.001
Valerate	42 ± 24	11 ± 2.1	8 ± 1.7	<0.001	<0.001
Isoacids ^1^	65 ± 27	37 ± 3.8	57 ± 11	0.174	0.249
Molar proportions (mol/mol SCFA)					
Acetate	0.56 ± 0.03	0.65 ± 0.03	0.7 ± 0.03	<0.001	<0.001
Propionate	0.34 ± 0.04	0.22 ± 0.03	0.17 ± 0.02	<0.001	<0.001
Butyrate	0.10 ± 0.02	0.12 ± 0.007	0.12 ± 0.01	0.001	0.001
Isob + Isov	0.005 ± 0.0006	0.005 ± 0.001	0.009 ± 0.002	0.003	<0.001
Valerate	0.006 ± 0.003	0.002 ± 0.0004	0.002 ± 0.0003	<0.001	<0.001
Isoacids ^1^	0.009 ± 0.004	0.007 ± 0.001	0.010 ± 0.001	0.532	0.095
Gas production (µmol/g)	14,795 ± 169	15,004 ± 696	14,394 ± 206	0.162	0.035
Methane production (µmol/g)	19.9 ± 1.13	19.9 ± 1.32	21.3 ± 0.36	0.964	0.175
Ammonia (mg N/L)	144 ± 12	153 ± 11	162 ± 8.8	0.027	<0.001

^1^ Isobutyrate, isovalerate and valerate.

**Table 4 animals-13-01102-t004:** Production of SCFA, molar proportions of SCFA, gas and methane production, and ammonia concentration after 24 h of the *in vitro* fermentation of cellulose by fecal inocula obtained from pigs under heat stress conditions and fed diets supplemented or not with betaine (5 g/kg) or zinc (0.120 g/kg). Means ± standard deviations are shown.

CELLULOSE	Treatment			*p*-Value	
	Control	Betaine	Zinc	Bet vs. Control	Zn vs. Control
SCFA production:					
Total SCFA (µmol/g)	823 ± 202	928 ± 118	1027 ± 231	0.313	0.087
Acetate	321 ± 60	497 ± 37	554 ± 120	0.272	<0.001
Propionate	174 ± 52	187 ± 40	205 ± 54	0.603	0.287
Butyrate	140 ± 33	133 ± 31	131 ± 33	0.341	0.334
Isob + Isov	76 ± 13	96 ± 20	134 ± 30	<0.001	<0.001
Valerate	18 ± 5.5	15 ± 3.4	4 ± 0.8	<0.001	<0.001
Isoacids ^1^	94 ± 12	94 ± 22	138 ± 29	0.269	<0.001
Molar proportions (mol/mol SCFA)					
Acetate	0.50 ± 0.01	0.56 ± 0.04	0.55 ± 0.01	<0.001	0.001
Propionate	0.18 ± 0.02	0.19 ± 0.02	0.18 ± 0.02	0.254	0.637
Butyrate	0.17 ± 0.01	0.13 ± 0.02	0.12 ± 0.01	0.012	0.002
Isob + Isov	0.104 ± 0.01	0.098 ± 0.02	0.14 ± 0.01	0.179	<0.001
Valerate	0.039 ± 0.019	0.016 ± 0.001	0.008 ± 0.004	<0.001	<0.001
Isoacids ^1^	0.14 ± 0.02	0.11 ± 0.02	0.15 ± 0.01	0.360	<0.001
Gas production (µmol/g)	3513 ± 254	3137 ± 279	3672 ± 199	0.151	0.236
Methane production (µmol/g)	11.8 ± 0.45	12.2 ± 1.84	13.1 ± 0.29	0.561	0.104
Ammonia (mg N/L)	252 ± 9.3	265 ± 1.6	265 ± 14	0.002	0.002

^1^ Isobutyrate, isovalerate and valerate.

**Table 5 animals-13-01102-t005:** Mean values of bacterial group abundance (log CFU equivalents/g fresh matter) determined by quantitative PCR in Iberian pigs (*n* = 8/diet) fed a standard diet supplemented or not with betaine (5 g/kg) or zinc (0.120 g/kg) under chronic heat stress conditions (30 °C, 28 d).

	Control	Betaine	Zn	SEM	Bet vs. Control	Zn vs. Control
Total bacteria	9.88	9.81	9.92	0.053	0.469	0.529
*Lactobacillus*	8.15	7.87	8.10	0.150	0.250	0.827
*Bifidobacterium*	8.19	7.93	8.38	0.082	0.007	0.175
Enterobacteriaceae	6.46	6.67	6.49	0.210	0.547	0.908
*Bacteroides*	8.93	9.24	9.03	0.112	0.103	0.630
*Clostridium* cluster IV	8.84	8.80	8.94	0.083	0.054	0.470
*Clostridium* cluster XIV	9.16	8.65	8.73	0.061	<0.001	0.004

## Data Availability

The data presented in this study are available upon request to the corresponding author.

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
