# Peer review of "The Effect of Supplementation with Betaine and Zinc on In Vitro Large Intestinal Fermentation in Iberian Pigs under Heat Stress"

_animals, 2023, doi:10.3390/ani13061102_

Round 1

Reviewer 1 Report

In this manuscript, Pardo et al. investigated the effects of supplementation of betaine or zinc on hindgut fermentation of pigs under heat stress (HS) using a in vitro fermentation system by pigs faecal inocula. Overall, this manuscript may help readers to better understand the influence of betaine or zinc on fermentation characteristics of different substrates and may provide new ways for nutrition intervention programs under pigs heat stress conditions. A few comments are needed to address before it can be accepted. 

M&M

Lines 92-94, the authors should clarify why they chose 0.120 g/kg Zn sulfate or 5 g/kg betaine, but not other addition amounts?

Change the current Table 1 to appendix Table S1 , and the current Table 7 to Table S2, as these two tables are not appropriate to take up too much space of the main text.

Results:

It does not make sense that the authors performed the experiments of dietary Betaine and Zn supplementation in a well-designed and intact experiments, but they showed the results and discussions for these two additives, separately. The authors should compare these two additives results in parallel. By doing so, the authors could tell the readers which one is better to be added in the diet to mitigate heat stress for pigs.

The authors should not use single sentence as a separate paragraph. Such as line 217, 218-219, and 252-254, etc.

Replace the bold fonts to non-bold fonts in the results section. Such as increased acetate in line 199, “acetate molar proportion” in line 201, etc.

The Tables 2-6 should be shown right after the second section of the results in line 282.

Discussion:

Once again, it does not make sense to use single sentence as a separate paragraph. Such as Lines 374-375.

The authors should add a subtitle when they discuss the results of Betaine, as the manuscirpt has a subtitle for Zn in lines 380-381.

Lines 332-336, the authors should discuss why Betaine supplementation increased the acetate production with the substrates of pectin or cellulose but not with the starch. Rather than to restate the current results again in the discussion section.

Line 400, change Starke to Starke et al.. Same for others such as Pieper line 402.

Author Response

Comments and Suggestions for Authors

In this manuscript, Pardo et al. investigated the effects of supplementation of betaine or zinc on hindgut fermentation of pigs under heat stress (HS) using a in vitro fermentation system by pig’s faecal inocula. Overall, this manuscript may help readers to better understand the influence of betaine or zinc on fermentation characteristics of different substrates and may provide new ways for nutrition intervention programs under pig’s heat stress conditions. A few comments are needed to address before it can be accepted. 

M&M

Lines 92-94, the authors should clarify why they chose 0.120 g/kg Zn sulfate or 5 g/kg betaine, but not other addition amounts?

Thanks for your comment.

Zinc supplementation dose was chosen as a concentration close to the maximum permitted dose rate of 150 ppm total dietary zinc at the European Union (See L380-383).

On the other hand, betaine concentration in the diet was the level at which the greatest benefits were obtained in a titration study (0, 0.125, 0.25, or 0.5% betaine) with growing pigs (Fernandez-Figares et al., 2002). This concentration has been used since 2002 in our lab as the preferred dose for betaine inclusion in pigs.

Fernández-Fígares I, Wray-Cahen D, Steele NC, Campbell RG, Hall DD, Virtanen E, Caperna TJ. Effect of dietary betaine on nutrient utilization and partitioning in the young growing feed-restricted pig. J Anim Sci. 2002 Feb;80(2):421-8. doi: 10.2527/2002.802421x.

Change the current “Table 1” to appendix “Table S1 ”, and the current “Table 7” to “Table S2”, as these two tables are not appropriate to take up too much space of the main text.

Thanks for your suggestion. We have made the changes and renumbered accordingly the rest of tables in the manuscript.

Results:

It does not make sense that the authors performed the experiments of dietary Betaine and Zn supplementation in a well-designed and intact experiments, but they showed the results and discussions for these two additives, separately. The authors should compare these two additives’ results in parallel. By doing so, the authors could tell the readers which one is better to be added in the diet to mitigate heat stress for pigs.

Thanks for your comment. In our opinion, comprehension is facilitated keeping the effects of betaine and zinc separated as they are. Our purpose was not to compare betaine and zinc effects on fermentation as it is known that their mechanisms of action differ. Alternatively, discussing the results separately facilitates the understanding of the role of betaine and zinc on fermentation. Additionally, it is not our intention to tell the reader, from an in vitro experiment as designed, which additive is better for heat stress.

The authors should not use single sentence as a separate paragraph. Such as line 217, 218-219, and 252-254, etc.

Thanks. We have avoided single sentences in separate paragraphs.

Replace the bold fonts to non-bold fonts in the results section. Such as “increased acetate” in line 199, “acetate molar proportion” in line 201, etc.

We have substituted bold font for regular font throughout the text.

The Tables 2-6 should be shown right after the second section of the results in line 282.

We have moved tables to the end of  the results section.

Discussion:

Once again, it does not make sense to use single sentence as a separate paragraph. Such as Lines 374-375.

Right. We have included the sentence in the previous paragraph.

The authors should add a subtitle when they discuss the results of Betaine, as the manuscirpt has a subtitle for Zn in lines 380-381.

Thanks for the comment. You are right. We have included a subtitle before discussing the results of betaine (L312-313).

Lines 332-336, the authors should discuss why Betaine supplementation increased the acetate production with the substrates of pectin or cellulose but not with the starch. Rather than to restate the current results again in the discussion section.

Thanks for your comment. As a matter of fact, although acetate production did not attain statistical significance when starch was fermented, a trend towards increased acetate molar proportion is now reported in the results section (L203-204). Although of less magnitude than with other substrates, there is an effect of betaine on acetate molar proportion when starch is fermented. However, the elevated variability of acetate production when starch was fermented has precluded to attain statistical significance.

Line 400, change “Starke” to “Starke et al.”. Same for others such as “Pieper” line 402.

Thanks for the correction.

Reviewer 2 Report

The manuscript investigates the effects of betaine or zinc supplementation and heat stress on changes in colonic fermentation. I congratulate the authors on their work, the experiment and results are interesting, and the manuscript is well written. Accordingly, I only have minor comments.  

My major concern for the manuscript is the statistical analysis section. The manuscript states that the results of the groups (control, betaine, Zn) were analysed by ANOVA then the results discriminated by a post-hoc Dunnett's test. I am not familiar with the outputs from SAS, but in my experience the results of dunnetts are not a P-value as listed in the tables. Further the results are given for the betaine/Zn vs control and not the ANOVA. In principal I have no concerns but could the authors expand the statistics section to provide more clarity about the testing to empower the reviewer to comprehend the results?

I found the paper easy to read, but would request the the authors reconsider the amount of superfluous information in the discussion. The section on Betaine runs from L284 to L379, 95 lines in total. The results of the experiment are not discussed until L328, 44 lines in and meaning that the authors are nearly half way through their discussion before switching to their results. The Zn section has similar proportions. Could the authors reconsider the amount of preamble that should belong in, or is repetitious of the introduction?

Continuing from above I think that the manuscript is lacking a discussion on the oral bioavailability of betaine and zinc and how this may influence the results. As the authors highlight HS changes ileal digestibility, meaning that the composition of digesta arriving at the colon may be different. There may be composition differences due to betaine or zinc ameliorating the impacts of HS. Or it may be direct actions in the GIT. In the human literature it appears that betaine has rapid absorption, meaning that there would be less betaine directly reaching the colonic microbiota. Zinc bioavailability differs substantially depending on the form, but I understand that ZnO has relatively low oral bioavailability and therefore would reach the colon. How would these different bioavailabilities influence the results?

Minor comments

L 53 what is meant by "scattered sweat glands?

L56 the authors state "intestinal integrity". Maybe be more specific as the study investigates colonic permeability, some other studies investigate the effects on the small intestine

All tables and figures have 4 decimal places for P-Values. Please limit this to 2 for values that are not significant (P>0.10) and 3 for those that are (P<0.10). This will reduce clutter and draw the eye to significant responses.

There are multiple instances where a comma is used in place of a decimal point in the tables. 

Author Response

Comments and Suggestions for Authors

The manuscript investigates the effects of betaine or zinc supplementation and heat stress on changes in colonic fermentation. I congratulate the authors on their work, the experiment and results are interesting, and the manuscript is well written. Accordingly, I only have minor comments.  

My major concern for the manuscript is the statistical analysis section. The manuscript states that the results of the groups (control, betaine, Zn) were analysed by ANOVA then the results discriminated by a post-hoc Dunnett's test. I am not familiar with the outputs from SAS, but in my experience the results of dunnetts are not a P-value as listed in the tables. Further the results are given for the betaine/Zn vs control and not the ANOVA. In principal I have no concerns but could the authors expand the statistics section to provide more clarity about the testing to empower the reviewer to comprehend the results?

Good catch! As a matter of fact, Dunnett´s was used in a previous version but we decided to carry out pre-planned contrasts after consulting a statistician in the final version. Sorry for the mistake. It now reads L188-189 “Additionally, pre-planned contrasts were generated using the contrast statement procedure of SAS to evaluate treatment effects (Control vs. Zn, and Control vs. Betaine). “

I found the paper easy to read, but would request the authors reconsider the amount of superfluous information in the discussion. The section on Betaine runs from L284 to L379, 95 lines in total. The results of the experiment are not discussed until L328, 44 lines in and meaning that the authors are nearly half way through their discussion before switching to their results. The Zn section has similar proportions. Could the authors reconsider the amount of preamble that should belong in, or is repetitious of the introduction?

Thanks for the input. Although it is true that we have devoted a relatively large foreword to introduce the reader before discussing our results, we think that this strategy prepares the reader and puts the results into context. We do not think that the discussion is redundant from what it was stated in the introduction.

Continuing from above I think that the manuscript is lacking a discussion on the oral bioavailability of betaine and zinc and how this may influence the results. As the authors highlight HS changes ileal digestibility, meaning that the composition of digesta arriving at the colon may be different. There may be composition differences due to betaine or zinc ameliorating the impacts of HS. Or it may be direct actions in the GIT. In the human literature it appears that betaine has rapid absorption, meaning that there would be less betaine directly reaching the colonic microbiota. Zinc bioavailability differs substantially depending on the form, but I understand that ZnO has relatively low oral bioavailability and therefore would reach the colon. How would these different bioavailabilities influence the results? 

This is an interesting observation. In the case of betaine, we chose to supplement a relatively high concentration in the diet (5 g/kg compared to 1.0 g/kg which is the most common dose in farms), which probably made a greater amount of betaine reach the large intestine.

In the case of zinc, the relatively low digestibility of inorganic sources actually poses an environmental risk as part of dietary zinc is excreted with the faeces. Of course, this inorganic zinc by passing the small intestine zinc may affect hindgut fermentation.

We have not measured zinc or betaine in the faeces. However, from the results it can be concluded that betaine and zinc do affect intestinal fermentation in the pig.

Minor comments

L 53 what is meant by "scattered sweat glands?

Scattered as in dispersed sweat glands (not a lot of them, diluted).

L56 the authors state "intestinal integrity". Maybe be more specific as the study investigates colonic permeability, some other studies investigate the effects on the small intestine

Thanks for the comment. The study by Pearce et al. (2013) showed decreased jejunum transepithelial electrical resistance, increased lipopolysaccharide (LPS) permeability and plasma endotoxin levels in heat stressed pigs. However, the authors summarized these effects as decreased intestinal integrity and that’s why we have kept such nomenclature.

Pearce, S. C.; Mani, V.; Weber, T.E.; Rhoads, R.P.; Patience, J.F.; Baumgard, L.H. Gabler, N.K. Heat stress and reduced plane of nutrition decreases intestinal integrity and function in pigs. J. Anim. Sci 2013, 91, 5183–5193. https://doi.org/10.2527/jas.2013-6759.

All tables and figures have 4 decimal places for P-Values. Please limit this to 2 for values that are not significant (P>0.10) and 3 for those that are (P<0.10). This will reduce clutter and draw the eye to significant responses.

We have changed the decimal places as requested.

There are multiple instances where a comma is used in place of a decimal point in the tables. 

We have exchanged commas for decimal points in the tables as requested.